# Neurophysiological Markers of Design-Induced Cognitive Changes: A Feasibility Study with Consumer-Grade Mobile EEG

**DOI:** 10.3390/brainsci15050432

**Published:** 2025-04-23

**Authors:** Nathalie Gerner, David Pickerle, Yvonne Höller, Arnulf Hartl

**Affiliations:** 1Institute of Ecomedicine, Paracelsus Medical University, 5020 Salzburg, Austriaarnulf.hartl@pmu.ac.at (A.H.); 2Institute for Diagnostic and Interventional Radiology, Favoriten Hospital, 1100 Vienna, Austria; 3Faculty of Psychology, University of Akureyri, 600 Akureyri, Iceland; yvonne@unak.is

**Keywords:** neuroarchitecture, evidence-based design, cognitive dynamics, attention restoration, EEG metrics, ecological validity

## Abstract

Background: Evidence-based design aims to create healthy environments grounded in scientific data, yet the influence of spatial qualities on cognitive processes remains underexplored. Advances in neuroscience offer promising tools to address this gap while meeting both scientific and practical demands. Consumer-grade mobile EEG devices are increasingly used; however, their lack of transparency complicates output interpretation. Well-established EEG indicators from cognitive neuroscience may offer a more accessible and interpretable alternative. Methods: This feasibility study explored the sensitivity of five established EEG power band ratios to cognitive shifts in response to subtle environmental design experiences. Twenty participants completed two crossover sessions in an office-like setting with nature-inspired versus urban-inspired design elements. Each session included controlled phases of focused on-screen cognitive task and off-screen breaks. Results: Factorial analyses revealed no significant interaction effects of cognitive state and environmental exposure on EEG outcomes. Nonetheless, frontal (θ/β) and frontocentral (β/[α + θ]) ratios showed distinct patterns across cognitive states, with more pronounced contrasts in the nature-inspired compared to the urban-inspired design conditions. Conversely, occipital ([θ + α]/β), (θ/α), and (β/α) ratios remained consistent across exposures. Data triangulation with autonomic nervous system responses and performance metrics supported these observations. Conclusions: The findings suggest that EEG power band ratios can capture brain–environment interactions. However, limitations of consumer-grade EEG devices challenge both scientific rigour and practical application. Refining methodological reliability could improve interpretability, supporting more transparent and robust data-driven design decisions.

## 1. Introduction

Building standards increasingly incorporate health-focused criteria to promote occupant well-being. A key strategy in this development is evidence-based design, which relies on data-driven research to guide design decisions that are both scientifically robust and practically applicable. Yet, while research informing building standards has traditionally emphasised factors that affect physical health [1], the influence of spatial qualities on psychological functioning and mental health remains relatively underexplored.

Neuroscience offers promising tools to bridge this gap by objectively measuring physiological responses to even subtle environmental experiences, thereby moving beyond subjective evaluations and strengthening evidence-based design strategies. Emerging fields such as neuroarchitecture [2], neurourbanism [3], and environmental neuroscience [4] are pioneering brain activity analyses to inform health-supportive design. However, because much of the current body of evidence is derived from lab studies with limited ecological validity [5], it is crucial to validate these findings in real-world settings to ensure that scientific insights can be effectively integrated with the practical demands for creativity, cost-effectiveness, and scalability in design.

Advances in electroencephalography (EEG) technology enable flexible, ecologically valid measurements of fast neural dynamics during environmental exposure. In particular, mobile EEG devices support continuous data acquisition while permitting natural movement [6,7]. Additionally, consumer-grade solutions are gaining popularity due to their availability, affordability, and ease of use—making data collection more accessible beyond specialised fields. Although a recent scoping review [8] projects their continued widespread adoption, validation efforts remain limited and concentrated on only a few models, fuelling ongoing debates about their internal validity and reliability compared to research-grade systems [5,9,10].

To date, most studies evaluating consumer-grade mobile solutions have primarily examined headsets, with a strong focus on Emotiv models. While these devices can reliably capture common brain activity changes [11,12,13,14], they also exhibit notable limitations. Hardware issues including low electrode density, inconsistent placement, poor skin contact, and battery decline [15,16,17] negatively impact data quality and contribute to overall higher noise levels and lower signal-to-noise ratios [18], compromising accuracy. Furthermore, while integrated software streamlines signal processing, reliance on proprietary “black box” algorithms for data analysis limits transparency. These algorithms convert raw signals into ready-to-use outputs ranging from power spectral density transformations to scaled indicators of mental states (e.g., stress, excitement, relaxation, engagement, interest, focus, meditation) as used in various environmental studies [19,20,21,22]. Although efficient, this approach limits statistical analysis, complicates interpretation, undermines validity [5], and hinders the translation of findings into evidence-based practices.

Two early studies illustrate these challenges in both laboratory [22] and outdoor settings [19], pairing Emotiv EEG metrics to assess emotional and cognitive dimensions of potential restorative processes induced by environmental exposure. One metric pair (“excitement” vs. “meditation”) aimed to capture emotional arousal, while the other (“engagement” vs. “frustration”) targeted cognitive interest. Arousal-related findings were consistent across both studies, but interest-related findings were contradictory. Due to opaque signal processing, neither researchers nor developers could explain these discrepancies, raising speculations about proprietary metrics not aligning with intended research objectives. Nonetheless, the use of paired EEG metrics marks a methodological advance, moving beyond simplistic positive-or-negative interpretations. This accommodates calls for a more nuanced understanding [5], as well as considerations of individual, situational, and contextual variability—insights critical for evidence-based design.

Cognitive neuroscience offers a blueprint for overcoming operational challenges. Neuroergonomics [23], which targets optimising human–system compatibility through brain activity analyses [24,25], shares the goal of enhancing person–environment fit. In this context, EEG metrics—particularly power band ratios, which compare relative power across frequency bands—are well established for monitoring cognitive dynamics [26]. These ratios reliably track changes associated with engagement [27,28,29], attentional control [30,31,32], mind-wandering [33], arousal [34,35], and fatigue [36,37,38,39]. Their computational efficiency and ease of interpretation make them promising tools for evaluating evidence-based design applications in complex, real-world contexts.

The present study examines whether such EEG power band ratios can serve as markers of cognitive dynamics in subtle human–environment interactions within an office-like setting. By comparing changes between screen-based tasks and off-screen micro-breaks, we aim to understand these dynamics through the lens of Attention Restoration Theory (ART) [40], which posits that sustained, effortful attention leads to cognitive fatigue, whereas restorative experiences—often facilitated by natural environmental features—engage effortless “soft fascination” that promotes mental recovery.

We hypothesise that selected EEG power band ratios (i) can be reliably measured in a controlled yet realistic setting using a validated mobile consumer-grade device with semi-automated signal processing, and (ii) are sensitive to brain activity changes induced by different interior designs. These hypotheses will be supported if the following occur:EEG metrics reveal a core pattern that distinguishes effortful on-task attentional effort from off-task default mode.EEG metrics capture nuanced modulations of this core pattern in response to varying environmental stimuli.EEG patterns align with autonomic nervous system responses and cognitive task performance metrics.

Assuming that attention restoration holds, we predict that environments featuring nature-inspired design elements will exhibit a more efficient balance of cognitive resource allocation (i.e., direction of attentional focus) and regulation (i.e., management of resource availability) compared to those with non-natural design elements.

## 2. Materials and Methods

We conducted a within-subject crossover study between July and November 2019 at the Paracelsus Medical University in Salzburg, Austria. Figure 1 illustrates the study design and procedural flow. Two groups of participants experienced the same office scenario twice, with subtle interior design changes following a G × U|U × G protocol: one group experienced condition G first, followed by condition U, while the other group underwent the reverse order. A 2–4-week washout period was implemented between sessions to minimise carryover effects. Sessions were scheduled consistently on the same time slots and weekdays.

### 2.1. Recruiting

Eligible participants were healthy adults aged 18–65 years with normal or corrected vision and intact colour perception. A total of 24 participants were recruited through a rolling enrolment process over the entire study period. They were randomly assigned to crossover groups in an alternating sequence, stratified by age and gender. Upon study completion, participants received a small token of appreciation (i.e., a set of soft drinks).

### 2.2. Experimental Setup and Environmental Stimuli

The study took place in a 20 m^2^ experimental room (3 m height) on the first floor of a building located off-campus. The room featured a minimalistic office design with white walls, white office cabinets at the back, and a white desk row with a black office chair in the centre. Two windows faced east (heavy-traffic street) and south (backyard) but remained shut with blinds drawn. While the windows were shut and the door closed, ambient noise remained uncontrolled as the room was not soundproof. Room lighting was controlled via eight LED tubes (150 cm, 27 W, 6500 K, 2700 lm). The room was ventilated for at least 10 min before each session, but temperature remained uncontrolled due to the lack of air conditioning. The technical setup (recording notebook, webcam, speakers) was placed in the back.

Participants’ visual field within the experimental setup, illustrated in Figure 2a, included two environmental stimuli: a neutral stimulus in the left periphery (L); and an experimental stimulus in the right periphery (R). The neutral stimulus, a blue pinboard, served to balance the visual field and minimise bias toward the experimental stimulus. Both peripheral stimuli were comparable in size (1 m^2^) and colour contrast, and were symmetrically positioned within participants’ visual field, mounted at 90/190 cm from the floor.

Figure 2b illustrates the two experimental stimuli altering between crossover sessions: a nature-inspired design element, featuring a four-level hydroponic vertical garden with 24 foliage plants, and a non-natural design element, featuring grey Styrofoam cuboids. Both stimuli were designed to be comparable in dimension, depth, and complexity to ensure consistent visual engagement. To promote realistic interaction, design changes were undisclosed.

### 2.3. Experimental Procedure

Crossover sessions followed the protocol outlined in Figure 3. Each session started with a 90 s fatigue induction phase, during which participants completed a cognitive loading task in paper-and-pencil format. This was followed by a block-design trial consisting of six alternating blocks of off-screen breaks (3 × 180 s) and computerised attention tasks (3 × 380 s). At the start of the first task block, participants completed a 120 s training block to practice the computerised task. The total duration was approximately 35 min.

Before each session, participants muted their phones and stored personal belongings out of sight. Following a briefing, they were fitted with wearable physiological sensors and seated at the desk in front of a closed notebook. The experimenter entered the room only to provide and collect the initial paper-and-pencil task but otherwise monitored the session remotely, giving verbal cues to signal the start and end of breaks. During task blocks, participants focused exclusively on the notebook screen. During off-screen breaks, they closed the notebook and remained seated to avoid interferences with the technical setup behind them. They were instructed to keep their eyes open but were not directed to look at any specific object, allowing their gaze to rest naturally within their visual field.

### 2.4. Data Acquisition

#### 2.4.1. Cognitive Tasks

To induce cognitive fatigue, we administered two parallel paper-based versions of the Digit Symbol Substitution Task (DSST) [41]. Participants matched symbols to numbers within a set time using a provided key. This task was used solely to standardise initial cognitive load across participants and sessions for comparability; therefore, task performance parameters were not analysed.

For focused task performance, we used a short version of the Attention Network Test (ANT) [42], presented on a 17.3-inch notebook (1600 × 900 resolution, 100% brightness) via Inquisit Lab (Version 5.0.14). ANT included a 24-trail training block with instructions and feedback, followed by three 96-trial experimental blocks. Each trial began with a fixation cross, followed by a cue and a stimulus—an array of five arrows, which had either congruent or incongruent orientations, positioned either above or below the fixation. Participants responded to the central arrow’s direction using arrow keys.

Overall cognitive performance was assessed through mean reaction times and accuracy. Attention network efficiency was assessed through mean reaction time differences: the alerting effect (cued vs. uncued trials, indicating alerting responsiveness), the orienting effect (trials with vs. without reliable spatial cues, indicating spatial responsiveness), and the conflict effect (congruent vs. incongruent flanker trials, indicating susceptibility to interference).

#### 2.4.2. Physiological Measures

Biometric data were acquired from wearable devices, and synchronised with iMotions software (Version 7.2, iMotions A/S, Copenhagen, Denmark).

Autonomic nervous system measures were recorded from the non-dominant hand using a 3-channel Shimmer3 GSR + unit (Shimmer Research Ltd., Dublin, Ireland) with a 128 Hz sampling rate. Heart activity was recorded via an optical pulse sensor on the index finger, and electrodermal activity was recorded via two Ag/AgCl electrodes on the middle and ring fingers. The output parameters were heart rate (HR) in beats per minute (bpm) and skin conductance (SC) in microSiemens (µS).

Brain activity was recorded using a wireless Emotiv Epoc + headset (Version 1.1, EMOTIV Inc., San Francisco, CA, USA) with 14 Ag/AgCl electrodes, fitted with saline-soaked felt pads. The one-size headset was placed on participants’ heads according to the manufacturer’s instructions [43], with the two electrode arms positioned over the left and right hemisphere to approximate the international 10–20 system (AF3, AF4, F7, F8, F3, F4, FC5, FC6, T7, T8, P7, P8, O1, O2), CMS/DLR references at M1/M2. Adjustments were made to ensure proper placement using landmarks as outlined in the user manual: reference sensors over the mastoid bones behind the earlobes, and front sensors approximately three finger-widths above the eyebrows, near the hairline. Signal quality was assessed via a sensor map in the Emotiv Cortex App (Version 1.3) and optimised through manual electrode adjustments and, when necessary, additional electrode moistening to ensure proper signal conduction. EEG signals were internally digitised at 2048 Hz (14-bit, 0.51 μV), down-sampled to 128 Hz, filtered (5th-order Sinc, a dual notch 50/60 Hz, bandpass 0.16–43 Hz), and automatically decomposed into theta (4–8 Hz), alpha (8–13 Hz), low beta (13–21 Hz), high beta (21–30 Hz), and gamma (30–45 Hz) bands. The output was the averaged absolute band power per sample (µV^2^/Hz) for each EEG channel and each spectral power band.

### 2.5. Data Processing

The synchronised data were video annotated in iMotions to mark trial block segments and subsequently processed in R (Version 4.3.1). Sampled data were screened for quality before averaging per trial block.

HR data showed 4.40% signal loss, primarily attributable to a single case. The dataset followed a consistent pattern (50–100 bpm), with occasional outliers (~120–150 bpm) that did not affect averages. SC metadata confirmed 99.98% signal validity with no losses. Case-by-case screening revealed no notable outliers or deviations.

EEG data showed no signal loss in the analysed segments, though one recording lacked the first rest block. Case-by-case screening identified extreme outliers, suggesting signal contamination that could bias analyses. Extracortical artefacts, such as eye blinks or muscle activity, are common in EEG and typically removed before processing. However, since we only had access to automatically pre-processed data—not the raw signal—we could not apply conventional methods for artefact detection and removal.

To assess irregularities in the pre-processed data, we first down-sampled it into 1 s and 10 s epochs. We then systematically screened both versions in parallel, iteratively applying conditional colour formatting across the 70 EEG variables (Appendix B, Figure A1). This allowed us to establish cut-off thresholds for spectral and temporal irregularities.

To recover analysable segments, the following additional processing steps were implemented: samples with values exceeding the cut-off thresholds were replaced by missing values across all power bands within the affected channel. This applied to values ≥ 5 μV^2^/Hz (equivalent to a conventional low-pass filter), and time segments with no signal changes over five seconds (indicating non-physiological flatlined data, often caused by poor electrode contact or amplifier saturation). In total, 29.52% of samples were removed due to potential signal contamination, with a similar proportion (25.27%) excluded when considering only the subset of channels ultimately retained for analysis. The remaining data points were averaged into 1 s epochs.

To further minimise the impact of signal contamination, prefrontal and distal channels (AF3, AF4, F7, F8, T7, T8, P7, P8) were excluded due to their heightened susceptibility to eye blink and muscle artefacts. Similarly, high beta and gamma bands were omitted, as they are particularly prone to contamination from muscle activity and amplifier noise. Detailed information on the average EEG spectral power for the channels and power bands included in our analysis, both before and after artefact removal, is available in Appendix B, Figure A2.

Five EEG power band ratios were computed for each channel within the designated regions of interest and subsequently averaged across all channels within the corresponding region. Regions of interest were defined based on their functional roles in cognitive processing: frontal (F3, F4) and frontocentral (FC5, FC6) sites for their involvement in higher-order cognition and sensory integration, and occipital (O1, O2) sites for their role in visual processing. Table 1 outlines the regions of interest and their channels, the computational formulas of the selected EEG metrics, along with observed dynamics in specific cognitive states, their interpretation within prior research contexts, and relevant references.

The frontal (θ/β) ratio, well investigated in cognitive neuroscience, serves as a dual marker of cognitive allocation. Lower levels indicate on-task attentional control and resilience to task-related distress [30,31,32], while higher levels link to internal thoughts off-task, such as mind-wandering [33]. The (β/[α + θ]) ratio is known as an index of on-task cognitive engagement, commonly used to assess complex tasks that require sustained attention, such as piloting [27,28,29]. Initially investigated over central and parietal regions, it has also been successfully measured in frontocentral regions [24], correlating with task load [44]. The occipital ([θ + α]/β) ratio serves as a marker for task-induced low-vigilance states, including visual fatigue. It has been applied in complex real-world settings such as driving, demonstrating correspondence with physiological and behavioural markers of reduced alertness and self-reported sleepiness [36,37,38,39]. Additionally, the occipital (β/α) has been associated with on-task arousal [34] and stress [35], inversely marking fatigue [37], and the occipital (θ/α) has been linked to visuospatial attention on task [45].

### 2.6. Statistical Analyses

Statistical analyses were conducted using R (Version 4.3.1). Independent crossover group differences were assessed using *t*-tests and Wilcoxon rank-sum tests for continuous variables, and Chi-Square tests for categorical variables.

Exploratory descriptive analyses across trial blocks and exposure conditions included summary statistics (means, standard deviations, 95% confidence intervals, medians, and interquartile ranges), and were visualised with line and violin plots to assess temporal activity patterns, data distribution, and outliers. Cases were excluded if (a) outliers skewed averages or (b) missing or outlier values affected more than one block per block type (task or rest) or entire EEG regions of interest. Single block averages were imputed within block type using the last observation carried forward and the next observation carried backward methods.

Comparative analyses were performed to examine exposure effects on dependent variables using the MANOVA.RM package (Version 0.5.4) [46] for semi-parametric ANOVA-type statistics. Two-factor analyses assessed the interaction and main effects of exposure (green vs. urban) and cognitive state (rest vs. focus) on continuously recorded physiological data, as well as the effects of exposure and time (1, 2, 3) on cognitive performance metrics. Given the small sample size, the results are reported using the Wald-type statistic (WTS) with resampled parametric bootstrap *p*-values and degrees of freedom from the central χ^2^ distribution [47].

For crossover group comparisons and post hoc analyses, paired *t*-tests were used for normally distributed data and Wilcoxon signed-rank tests for non-normal data, with normality assessed via the Shapiro–Wilk test. The significance level was set at α = 0.05 without corrections for multiple testing. After correction for multiple comparisons, the significance level was α/12 = 0.0042. This correction includes all hypothesis-relevant ANOVAs but not crossover group comparisons that aimed at comparing differences by study design, where not correcting represents the more conservative approach.

## 3. Results

### 3.1. Participant Information and Crossover Group Differences

Twenty-four participants were enrolled, with four lost to follow-up, resulting in a final sample of 20 (10 female), aged 20–66 years (M = 37.95, SD = 14.41). The washout period averaged 19.60 days (SD = 6.72), and session start times differed by 80.80 min on average (SD = 66.74). In total, 78% of sessions took place in the warm season (July–September). Table 2 details the crossover group differences. Groups were generally homogenous but differed significantly in ANT alerting effect and EEG frontal (θ/β) ratio. Additional analyses are provided in Appendix B, including group differences in ANT performance parameters after case exclusion (Table A1), as well as differences in average EEG spectral power across power bands and channels for the complete dataset (Table A2) and the analysed subset after case exclusion (Table A3).

### 3.2. Cognitive Performance

We identified one case with consistently high reaction times and low response accuracy in block 1 of the green condition. Results reported all data, with changes due to exclusion noted as they arose.

#### 3.2.1. Reaction Time

All participants had similar mean reaction times across both trials, with slightly higher values in the green condition (M = 459.36, SD = 58.74) than the urban condition (M = 457.49, SD = 58.90), most prominently in trial block 1 (see Figure 4a), with similar distribution across blocks and trials (see Figure 4b). A two-way ANOVA revealed no significant interaction effect between exposure and time, nor any main effect.

#### 3.2.2. Accuracy

Participants also showed a similar mean percentage of correct responses across trials, with slightly lower accuracy in the green condition (M = 97.64, SD = 3.12) than the urban condition (M = 98.19, SD = 1.70), which was present in the first two blocks but reversed in block 3 (see Figure 5a), most prominently in the identified outlier case’s green trial (see Figure 5b). A two-way ANOVA showed no significant interaction effect between exposure and time, but there was a significant main effect of time (*p* = 0.044, df = 2, W = 7.356), which vanished after outlier case exclusion and was not significant after correcting for multiple comparisons.

#### 3.2.3. Alerting Effect

The mean alerting effect was lower in the green condition (M = 46.14, SD = 22.20) compared to the urban conditions (M = 51.35, SD = 24.61). This pattern was evident in blocks 1 and 2 but reversed in block 3 (see Figure 6a), without notable deviations across blocks and trials (see Figure 6b). A two-way ANOVA revealed a tendency for a significant interaction effect between exposure and time (*p* = 0.060, df = 2, W = 6.401) and a main effect of time (*p* = 0.009, df = 2, W = 12.925), which was significant before but not after correcting for multiple comparisons. After case exclusion, the interaction effect disappeared, while the main effect persisted. Post hoc paired *t*-test showed a significant difference between the green condition (M = 35.30, SD = 22.45) and the urban condition (M = 49.46, SD = 24.01) in block 1 (t = −2.510, *p* = 0.021), which was also persistent after case exclusion.

#### 3.2.4. Orienting Effect

The mean orienting effect was higher in the green condition (M = 19.56, SD = 17.78) than in the urban condition (M = 14.85, SD = 15.23), with a consistent pattern across all blocks (see Figure 7a) and no notable anomalies (see Figure 7b). A two-way ANOVA showed no significant interaction effect between exposure and time. There was a tendency for a significant main effect of time (*p* = 0.058, df = 2, W = 7.273), which further diminished after case exclusion. Post hoc comparison showed a significant difference between green (M = 22.45, SD = 29.50) and urban (M = 12.56, SD = 33.18) conditions in block 2 (V = 160, *p* = 0.040), which was persistent after case exclusion.

#### 3.2.5. Conflict Effect

The mean conflict effect was higher in the green condition (M = 76.60, SD = 28.07) than in the urban condition (M = 75.08, SD = 21.13), a difference only observable in block 1 (see Figure 8a). No notable anomalies were detected across blocks, except for consistently higher values in the outlier case in the green condition (see Figure 8b). A two-way ANOVA revealed no significant interaction effect, nor any main effects. Excluding the outlier case did not alter statistical results, however, it reversed the means, with the green condition (M = 72.38, SD = 21.33) slightly lower than the urban condition (M = 72.91, SD = 19.29).

### 3.3. Autonomic Nervous System Response

Cases were excluded if required to facilitate comparative analysis; otherwise, the results reported all data, with changes due to exclusion noted as they arose.

#### 3.3.1. Heart Rate

One case with 93.20% signal loss and 5.20% outlier rate was excluded, leaving data from 19 participants. Mean HR was slightly lower in the green condition (M = 70.5, SD = 9.98) than in the urban condition (M = 71.1, SD = 10.90), with consistent activity pattern regardless of cognitive state, except block 1 where the pattern was reversed (see Figure 9a). Unique values were clustered within similar ranges in both conditions (see Figure 9b). A two-way ANOVA found no statistically significant interaction effects between exposure and cognitive state on mean HR, nor any main effects.

#### 3.3.2. Skin Conductance

Participants (N = 20) exhibited lower mean SC levels in the green condition (M = 5.56, SD = 7.94) than in the urban condition (M = 6.59, SD = 8.08) with consistent patterns of increase in both conditions regardless of cognitive state (see Figure 10a). Few outliers (see Figure 10b) originating from two cases skewed the data but did not affect the observed trends. A two-factorial ANOVA revealed no statistically significant interaction effects between exposure and cognitive state, nor any main effects. The exclusion of outlier cases did not alter the results.

### 3.4. Neurophysiological Response

Cases were excluded across regions of interest as needed to facilitate comparative analysis. Information on average EEG spectral power across the considered power bands and channels, before and after case exclusion for both the nature-inspired (green) and urban-inspired (urban) design conditions, is presented in Appendix B, Figure A3.

#### 3.4.1. Frontal (θ/β) Ratio

Single missing block averages were imputed in two cases. Two cases were excluded—one for insufficient data, another for a prominent outlier and reversed activity pattern—leaving 18 participants. The mean (θ/β) ratio was higher in the green (M = 3.66, SD = 1.45) than urban condition (M = 3.33, SD = 1.17). The mean ratios were consistently higher at rest than during tasks, overall increased and with steeper slopes in the green condition (see Figure 11a). Confidence intervals overlapped across trial blocks. Single observations did not vary between conditions (see Figure 11b). A two-way ANOVA showed no significant interaction between exposure and cognitive state but a main effect of cognitive state (*p* = 0.007, df = 1, W = 10.296), which was significant before but not after multiple comparisons.

#### 3.4.2. Frontocentral (β/[α + θ]) Ratio

Single missing trial block values were imputed in three cases. Two further cases were excluded due to extensive missing data, leaving data from 18 participants. The mean (β/[α + θ]) ratio was nearly identical in both green (M = 0.29, SD = 0.10) and urban conditions (M = 0.29, SD = 0.07). Figure 12a shows consistently higher mean ratios during task performance than rest, with stronger contrast between restful and engaged states in the green compared to the urban condition. Confidence intervals overlapped across trial blocks. The distribution of data points was consistent across conditions (see Figure 12b). A two-way ANOVA revealed no significant interaction between exposure and cognitive state but a main effect of cognitive state (*p* = 0.008, df = 1, W = 8.493), which was significant before but not after multiple comparisons.

#### 3.4.3. Occipital ([θ + α]/β) Ratio

In one case, an outlier block average was identified and imputed, as was a missing block average in two cases. Four cases were excluded due to insufficient data, leaving data from 16 participants. The mean ([θ + α]/β) ratio was slightly higher in the green condition (M = 4.28, SD = 1.34) than in the urban condition (M = 4.27, SD = 1.22). The activity pattern showed higher mean ratios during resting than task performance. Small differences between conditions were observable during task performance but not rest, notably in task block 2. Confidence intervals overlapped across trial blocks (see Figure 13a) and single observations did not vary between conditions (see Figure 13b). A two-way ANOVA showed no significant interaction effect between exposure and cognitive state but found a main effect of cognitive state (*p* = 0.044, df = 1, W = 4.82), which was significant before but not after multiple comparisons.

#### 3.4.4. Occipital (β/α) Ratio

A missing block average was imputed in two cases. Four cases were excluded due to insufficient data, leaving data from 16 participants. The mean (β/α) ratio was slightly lower in the green condition (M = 0.75, SD = 0.21) than in the urban condition (M = 0.76, SD = 0.23). Activity patterns were consistent during rest but varied during task performance, with trends showing lower mean ratios during rest and higher during task performance, especially in the green condition (see Figure 14a). Confidence intervals overlapped throughout trial blocks. The distribution of data points did not vary between conditions (see Figure 14b). A two-way ANOVA showed no significant interaction between exposure and cognitive state, nor any main effects.

#### 3.4.5. Occipital (θ/α) Ratio

A missing block average was imputed in two cases. After excluding four cases due to insufficient data, data from 16 participants remained. The mean (θ/α) ratio was slightly higher in the green condition (M = 1.31, SD = 0.41) than in the urban condition (M = 1.29, SD = 0.38). Activity patterns were consistent between conditions but reversed in task block 3 (see Figure 15a). The mean ratios alternated with cognitive states, increasing during rest and decreasing during task performance in both conditions, with overlapping confidence intervals across trial blocks. The distribution of data points did not notably vary between conditions (see Figure 15b). A two-way ANOVA showed no significant interaction between exposure and cognitive state but a main effect of cognitive state (*p* = 0.048, df = 1, W = 4.548), which was significant before but not after multiple comparisons.

## 4. Discussion

This study investigated whether EEG power band ratios, established in cognitive neuroscience, can serve as reliable indicators of human–environment compatibility in evidence-based design. We hypothesised that selected EEG metrics could be reliably recorded with a validated consumer-grade mobile EEG within a controlled yet realistic office-like setting and could indicate cognitive changes corresponding to different interior designs.

All EEG metrics displayed the expected core pattern, consistently distinguishing effortful attention from default mode processes in line with previous research. Subtle modulations in response to environmental stimuli were particularly noted in frontal and frontocentral regions, although no statistically significant differences were found. Findings from autonomic nervous system measures and cognitive performance metrics aligned with the EEG results, also lacking significant effects linked to environmental conditions.

Observed differences in ANT metrics were temporally limited to the first and second task blocks, affecting the alerting and orienting networks but not the executive network. This contrasts with ART, which posits that the restorative effects on directed attention would primarily influence top-down inhibitory processes [40,48], reflected in the conflict effect metric [42,49]. While data anomalies—including outlier effects, crossover group differences (higher levels in G × U compared to U × G), and habituation across trials—complicate interpretation, our findings align with a recent synthesis of brain imaging studies. These studies suggest that environmental stimuli, particularly nature compared to urban elements, primarily activate bottom-up attentional processes involved in basic visual processing and sensory integration, such as alerting and orienting [5]. In this context, the temporarily higher alerting efficiency in the urban condition may be due to greater sensory stimulation and heightened arousal, while the higher orienting efficiency in the green condition may reflect enhanced visuospatial processing. These interpretations are further supported by our findings of lower HR and SC in the green condition, indicating a calming effect of nature-inspired design, which is consistent with previous research on reduced autonomic activity during short-term exposure to indoor plants [50]. Such effects have also been linked to enhanced cognitive performance [51] and increased EEG alpha power, indicative of relaxed wakefulness [52]. Together, these trends provide a meaningful context for interpreting our EEG findings.

The frontal (θ/β) ratio revealed a consistent pattern—higher during rest and lower during tasks—with overall higher levels in the green condition, particularly during early rest phases. Although crossover group differences were observed (lower levels in G × U compared to U × G), consistent task–rest dynamics and similar responses to interior stimuli across groups suggest that these anomalies did not influence these findings. Backed by the existing body of literature, the overall high frontal (θ/β) activity indicates reduced top-down inhibitory attentional control [31,33], which may limit the capacity to suppress distractors but also promote greater internal allocation.

The frontocentral (β/[α + θ]) ratio, an established index of task-related cognitive engagement [27,28,29], was higher during tasks compared to rest, with more pronounced alternations in the green condition. This pattern suggests greater on-task engagement and more effective off-task disengagement in nature-inspired compared to urban-inspired interiors, reflecting flexible cognitive resource allocation that may facilitate more efficient resource regulation.

The occipital ([θ + α]/β) ratio, a marker of task-induced low-vigilance states [36,37,38,39], was higher during rest compared to task performance, which is consistent with the expected cognitive down-regulation patterns. Differences between the interior conditions were subtle, inconsistent, and mainly observable in on-task phases, suggesting similar cognitive resource availability across conditions, with some potential sensitivity to cognitive effort.

To further explore cognitive regulation, we analysed the occipital (θ/α) and (β/α) ratios separately. Both showed the expected inverse patterns, with higher (θ/α) and lower (β/α) ratios during rest, and the reverse during task performance. Again, differences between interior conditions were subtle and inconsistent, supporting our interpretation of similar cognitive resource recruitment. Slightly more pronounced patterns in the green condition during rest may suggest enhanced cognitive down-regulation, though this remains speculative. Previous research linked such resting-state increases in alpha and beta activity (with relatively lower beta) to greater subjective restorativeness and reduced stress in environments with moderate vegetation density [53], as incorporated in our study. Similarly, both occipital alpha and theta were linked to attention restoration off-task [54].

When considered alongside a temporarily less activated ANT alerting network, generally lower HR and SC, as well as more pronounced frontal (θ/β) internal allocation and frontocentral (β/[α + θ]) off-task disengagement, these occipital patterns become more indicative of cognitive down-regulation mechanisms. The nature of this down-regulation processes can vary, often interpreted as visual fatigue during active tasks but taking on different qualities in idle states. Specifically, a lower (β/α) ratio may reflect greater relaxation relative to arousal, and a higher (θ/α) ratio could signify deeper restorative down-regulation beyond relaxed wakefulness. These nuanced aspects warrant further investigation.

In summary, data triangulation suggests greater on-task engagement but lower attentional control, as well as greater off-task disengagement and internal focus in nature-inspired compared to urban-inspired interiors. These cognitive allocation patterns occurred under comparable resource regulation dynamics across conditions, with slightly more off-task down-regulation in the green condition. While lower top-down inhibitory attentional control might hinder tasks requiring sustained focus, it may benefit creative and problem-solving tasks. Thus, the frontal and frontocentral metrics could inform context-specific design decisions. Additionally, these metrics may also strengthen the empirical basis for such theoretical constructs like Kaplan’s ART, which proposes that nature stimuli support cognitive recovery as they demand less cognitive resources [40,55], and that such restorative processes also involve introspective reflection [40] (p. 172). In contrast, occipital metrics were less sensitive to environmental influences. Although subtle nuances were observed, their inconsistency limits interpretability.

Finally, the absence of significant environmental effects may stem from several factors. A common explanation is that the cognitive task did not sufficiently deplete participants’ cognitive resources, thereby rendering processes of cognitive recovery—central to ART—less relevant and limiting the potential to detect restorative effects. While the ART-based notion of recovery remains debated, as cognitive changes following environmental exposure may also result from mechanisms beyond recovery, such as cognitive enhancement and motivation [48,56], this interpretation aligns with previous research using more demanding sustained attention tasks than ANT, under similar designs [57]. Kaplan’s [40] concept of restorative dynamics further supports this view, distinguishing between a fast component, characterised by an immediate stress response and short-term recovery, and a slow component, involving gradual cognitive depletion and more durable recovery. Since ANT may not have sufficiently engaged these slower mechanisms, they were likely not captured in our data. In contrast, faster processes may have been more accurately reflected in autonomic nervous system measures, offering a valuable reference point for interpreting EEG signals. Nonetheless, given the high temporal resolution of EEG, we expected it to capture these faster dynamics as well. Notably, the established EEG metrics selected for frontal and frontocentral sites have previously shown being responsive to attentional control and engagement—processes more indicative of cognitive enhancement and motivational components—whereas occipital metrics were shown to be responsive to vigilance states, such as fatigue, which are often interpreted within the recovery framework. While frontal and frontocentral metrics may have reflected this sensitivity in our study, concerns regarding data validity remain and are discussed in the following section.

### 4.1. Challenges, Strengths, and Limitations

Our study sought to provide a simple and accessible alternative to proprietary EEG algorithms provided with consumer-grade devices, which limit interpretability and hinder research translation. While these devices do not meet research-grade standards, their accessibility makes them valuable for evidence-based design, where balancing scientific rigour with real-world constraints is essential. Easily accessible neurophysiological data can be particularly relevant during planning and design stages, and for pre- and post-occupancy evaluations. We opted for an approach that would balance these often conflicting demands of evidence-based design and neuroscience in applied settings, contributing to better reflection on the practical challenges, strengths, and limitations.

Methodologically, we aimed to strike a balance between controlled conditions and a naturalistic setting, combining elements of both to enhance the ecological relevance of our study. The crossover design with a washout period controlled for sequence and carryover effects, enhancing internal validity. Each participant served as their own control, reducing inter-individual variability and minimising sample size needs. Participants were randomly assigned to groups, balanced by age and gender—two variables shown to influence responses to plants [58]—allowing for a more robust comparison. The design enabled detailed analysis of exposure effects while accounting for potential habituation.

We prioritised economic validity by ensuring immersive, realistic experiences, allowing participants to shift their gaze freely during off-screen breaks without specific stimulus instructions. This subtle, non-directed exposure reflects real-world use and aligns with practical applications in design settings. The study also combined physiological and behavioural measures, adding depth to the interpretation of environmental impacts.

Despite several methodological advantages, a number of challenges and limitations emerged during the study. First, the dropout rate of 16.67% reduced the final sample size, which in turn limited statistical power. Although appropriate statistical methods for small samples were applied, the findings should be interpreted with caution. Another limitation of the crossover design was its reduced suitability when the mere perception of environmental stimuli could elicit responses or introduce bias based on expectations. To mitigate such bias, design changes were not disclosed to participants, although they were informed of the study’s general objective during enrolment. While participants were not explicitly directed to focus on the environmental stimuli, changes across sessions were not entirely unnoticed. Future studies could enhance robustness by adopting a between-subject design with parallel groups—ideally including a neutral stimulus condition—which could help minimise expectance effects and reduce the likelihood of participants detecting design changes. Finally, while a crossover study design is effective for isolating specific health effects, it may be less appropriate for studies in architectural or environmental research.

In addition to these methodological challenges, several factors related to ecological and internal validity need to be addressed. Despite our efforts to minimise environmental confounders, we were unable to control certain ambient factors, such as temperature and noise, due to structural limitations. These uncontrolled environmental variables may have influenced cognitive performance and EEG activity [59,60], potentially introducing confounds into the data. Moreover, while intra-individual differences were controlled through study design (e.g., by standardising baseline cognitive load across sessions), inter-individual differences were less tightly managed. We stratified groups by gender and age and excluded individuals with colour vision deficiency to enhance between-group comparability. However, we did not systematically assess other participant characteristics that could have influenced cognitive performance or EEG activity, such as educational background, cognitive or affective disorders, or aesthetic preferences—and we acknowledge their potential impact on the results.

One specific area of interest is the group differences observed in the ANT alerting effect and the frontal (θ/β) ratio, which could not be sufficiently clarified. Excluding a case with notably low response accuracy in the cognitive task did not affect these group-level differences (see Appendix B, Table A1), leaving no clear indicator for further explanation beyond general effects such as expectation or habituation. In contrast, group differences observed in the EEG data—specifically in the low beta band—were partially influenced by exclusions based on data quality (see Appendix B, Table A2 and Table A3). These EEG exclusions were unrelated to ANT performance and involved a larger number of cases, suggesting that signal contamination may have contributed to those effects.

It is important to note that this aspect of inter-individual differences also reflects differing priorities across research fields. While health research primarily focuses on homogenous groups to control variability, design research must address the reality of diverse occupant populations and develop inclusive solutions for heterogeneous contexts. In our study, we recruited healthy adult participants and excluded only those whose health conditions could have interfered with participation. However, even this minimal screening—such as the exclusion of individuals with colour vision deficiency—introduced a degree of selectivity from an ecological validity perspective. This exclusion is notable because colour vision deficiency has a global prevalence of about 8% [61] and does not typically impede office work—the setting we simulated in our study. Moreover, colour is just one aspect of design, and the objective measures examined in our study should ideally apply beyond colour-sensitive scenarios or aesthetic preferences, which are sometimes too narrowly equated with psychological well-being. Nonetheless, to build robust evidence for design interventions, inter-individual differences in responses must be considered—particularly when targeting specific design contexts.

To address this important aspect, future studies could include individual difference variables alongside EEG measures to better understand how neurophysiological responses may vary. A very common approach in design evaluations is the use of subjective reports, which can complement EEG and other physiological data. For example, questionnaires can be used to assess participants’ psychological states. However, care must be taken not to conflate psychological responses with design preferences, which carry their own confounding variables—one reason we focused on objective measures in our study. In addition, EEG metrics can be complemented by behavioural measures of human–environment interaction, such as eye-tracking. Overall, multi-method approaches can enrich the interpretation of EEG data introduced in design research.

Specifically for EEG applications, accounting for individual differences could benefit from baseline recordings to capture natural intra-individual fluctuations. Furthermore, defined frequency bands on an individual basis could help control for inter-individual variability. While the nature of the EEG output from the consumer-grade device used in our study did not permit customised frequency definitions, future research could explore EEG metrics within individualised spectral ranges. Admittedly, such signal transformations require access to raw data and technical expertise in EEG processing—factors that often run counter to the appeal of consumer-grade EEG. Nonetheless, future research could explore accessible methods for integrating individualised frequency band approaches into neuroarchitecture studies, particularly when higher-grade equipment and raw data are available.

Not least among the challenges were technical limitations. We used wearable devices to enable a more natural human–environment interaction. While the research-grade device for monitoring autonomic nervous activity performed reliably and produced high-quality data, the consumer-grade EEG device—central to our study—posed significant technical difficulties, which is consistent with previous findings. Despite taking precautions (e.g., raising participant awareness of artefact-inducing behaviours, performing signal quality checks before data acquisition, and using a motion-minimising setup), substantial signal contamination was observed in the EEG output, indicating absent or insufficient real-time artefact removal. While artefact sources could not be identified retrospectively, the issues we encountered align with common limitations reported in prior research, such as poor electrode contact due to the one-size headset design [17], interference from thick hair, and signal degradation over time, likely linked to battery performance [15]. In addition, topographical accuracy was limited: although electrodes followed the 10/20 system, precise positioning was not possible due to the fixed electrodes on the headset arms. Temporal accuracy also suffered due to real-time processing delays and synchronisation software limitations, which prevented accurate segmentation via video annotations. The use of real-time markers could have ensured greater consistency. Furthermore, as raw EEG data were not available, the contaminated signals had to be cleaned using non-standard methods, diverging from established neuroscience practices. This limitation also led to restricting the EEG ratio computations to the low beta band, whereas prior studies included the full beta range—potentially flattening activity patterns and limiting the interpretability of our findings.

Despite these challenges, our study represents a meaningful step toward making neurophysiological data accessible for evidence-based design in real-world settings. We recommend that future research prioritise improvements in data quality and ensure access to raw data for more detailed analysis. Although consumer-grade EEG promises user-friendly recording and processing, its performance fell short of expectations in our study. Headset sensor preparation was relatively time-intensive, as were the corrections needed to achieve acceptable signal conduction. Similarly, data processing required extensive quality checks and adjustment calls for cleaning already pre-processed data—demanding considerable time and resources. Given the degree of data loss and high effort involved, investing in mobile equipment that allows access to raw, high-quality EEG data may be a more efficient and sustainable solution for future research.

### 4.2. Implications for Research and Design Practice

Our study highlights the potential of EEG for monitoring cognitive states during passive human–environment interactions. Although most differences between design elements were not statistically significant, and those which were significant did not survive the conservative correction for multiple comparisons, this exploratory investigation offers valuable insights for research and design practice.

A key contribution is the introduction of well-documented, publicly available EEG metrics that can be easily applied to pre-processed power band data. These metrics were sensitive to subtle environmental stimuli and interpretable alongside cognitive and autonomic nervous system measures, providing a transparent and practical framework for future studies. This enables architects and designers to make informed, evidence-based decisions.

Future research can build on this by refining the use of EEG power band ratios in environmental contexts. Our data offer a foundation for estimating appropriate sample sizes and conducting power analyses. However, the challenges in post-processing and interpretation underscore the need for caution when using consumer-grade devices, especially where signal quality cannot be controlled. While our troubleshooting strategies helped mitigate data loss, access to high signal-to-noise ratio raw data remains essential. This access allows the implementation of individually defined power band ranges, which may not only improve accuracy but also enhance the generalisability of the findings.

Beyond research, our findings inform ergonomic design strategies for healthier buildings. Environmental impacts on well-being are highly context-dependent and shaped by individual susceptibility. Berman et al. [4] caution against overgeneralising neuroimaging results, instead envisioning spaces that dynamically adjust environmental parameters based on occupants’ brain activity. Relative EEG spectral power and individually defined ranges could help enable real-time detection of cognitive state changes, paving the way for adaptive environments. Building on the concept of existing feedback loops (e.g., adaptive task allocation with the EEG engagement index [29]), the integration of EEG ratios into design evaluation could support the creation of spaces that respond flexibly to shifting cognitive and emotional needs.

Specifically, EEG-based neurofeedback can enrich design processes and post-occupancy evaluations, which currently rely heavily on subjective self-reports and behavioural observations. In contrast, objective and transparent EEG metrics offer a complementary approach, reducing expectation and response biases while providing deeper insights into occupant needs. This is particularly relevant for inclusive design, where understanding neurodiverse and vulnerable populations is key. EEG signals, which may deviate from typical patterns in these groups, can reveal how design influences are experienced differently—ultimately fostering environments that support a broader range of cognitive and emotional states.

Finally, our findings contribute to the ongoing development of qualitative design approaches such as biophilic design, which faces challenges in standardisation due to the contextual and diverse nature of elements like plants, water features, or organic forms. These design elements are often difficult to quantify, and overly rigid guidelines risk overlooking their nuanced effects. The EEG metrics explored in our study offer objective insights into the neurophysiological dynamics that may underlie resource-efficient or restorative processes, such as shifts from effortful to effortless attention, described in ART as “soft fascination”. Additionally, they can help to clarify whether other mechanisms, including cognitive enhancement, increased motivation, sustained vigilance, or aesthetic preference, are at play. Overall, this approach offers a valuable operationalisation of debated concepts [56], enriching theoretical models in environmental psychology.

## 5. Conclusions

This study proposed an approach that integrates neuroscientific methods while balancing scientific rigour with ecological validity to generate actionable evidence for designing health-supportive environments.

A key contribution of this work is the introduction of transparent EEG markers based on well-established power band ratios to objectively measure subtle human–environment interactions. Unlike proprietary EEG metrics with opaque algorithms, these markers ensure that the findings are interpretable and applicable. Exploratory analyses identified EEG patterns indicative of cognitive resource allocation and regulation, supported by triangulation with autonomic nervous system and cognitive performance data. The study also highlights persistent challenges with low-cost consumer-grade EEG devices, such as poor signal quality and restricted access to raw data, which limited their role for economic validity. Troubleshooting measures were introduced to manage data contamination retrospectively, though this cannot replace high-quality data from the outset.

By equipping researchers and designers with practical tools for incorporating EEG-based insights into environmental design, this work lays the foundation for future advancements. Addressing technological limitations and refining these methods will be critical for broader application, ensuring context-specific generalizability and promoting informed, health-supportive design practices.

## Figures and Tables

**Figure 1 brainsci-15-00432-f001:**
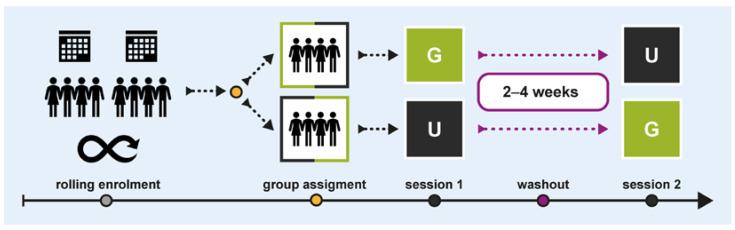
Study design and procedural flow.

**Figure 2 brainsci-15-00432-f002:**
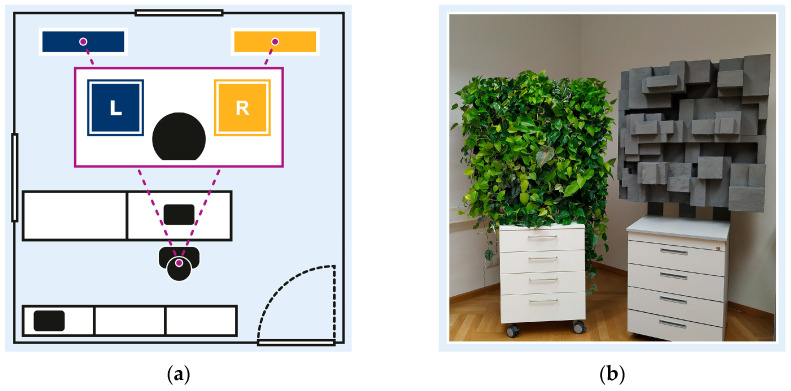
Experimental setting: (**a**) experimental room and participants’ visual field, with neutral stimulus in left periphery (L), and experimental stimulus in right periphery (R); (**b**) experimental stimuli including nature-inspired design element (left) and urban-inspired design element (right).

**Figure 3 brainsci-15-00432-f003:**
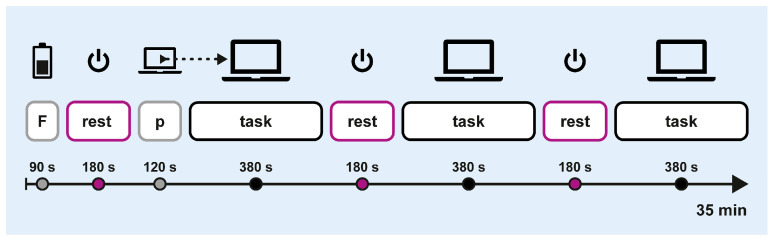
Experimental procedure. F: fatigue induction; p: practice.

**Figure 4 brainsci-15-00432-f004:**
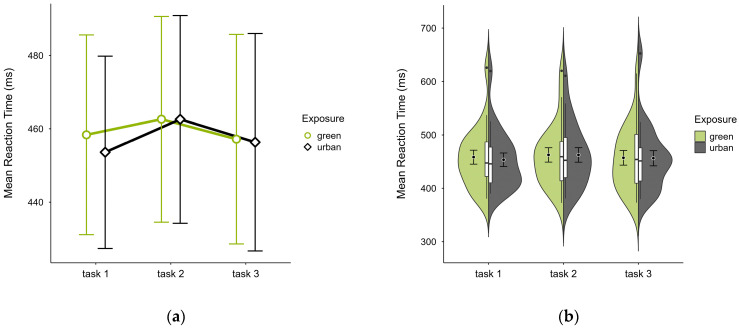
ANT reaction time (ms) across task blocks and design conditions (green vs. urban): (**a**) mean changes over time with CIs; (**b**) distribution and variability of observations across task blocks, medians with IQRs (centre), means, and CIs (periphery).

**Figure 5 brainsci-15-00432-f005:**
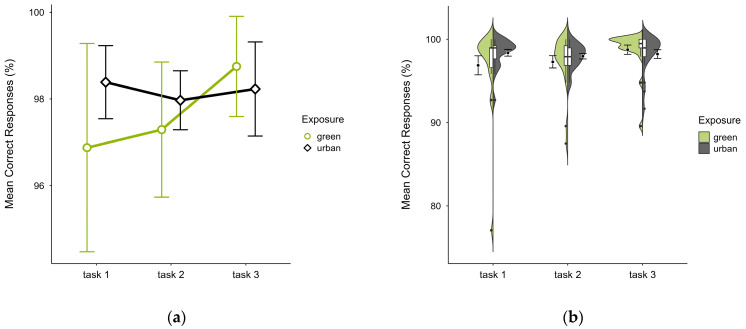
ANT correct responses (%) across task blocks and design conditions (green vs. urban): (**a**) mean changes over time with CIs; (**b**) distribution and variability of observations across task blocks, medians with IQRs (centre), means, and CIs (periphery).

**Figure 6 brainsci-15-00432-f006:**
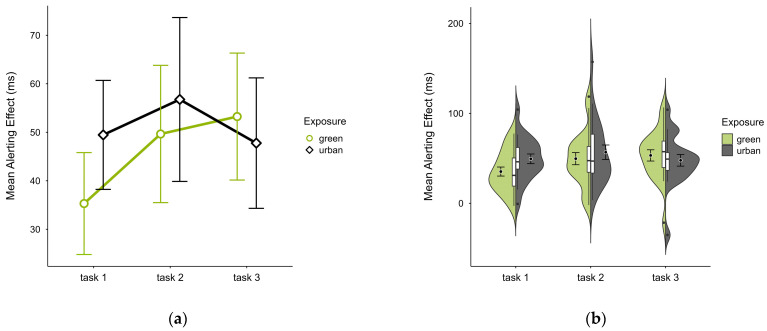
ANT alerting effect (ms) across task blocks and design conditions (green vs. urban): (**a**) mean changes over time with CIs; (**b**) distribution and variability of observations across task blocks, medians with IQRs (centre), means, and CIs (periphery).

**Figure 7 brainsci-15-00432-f007:**
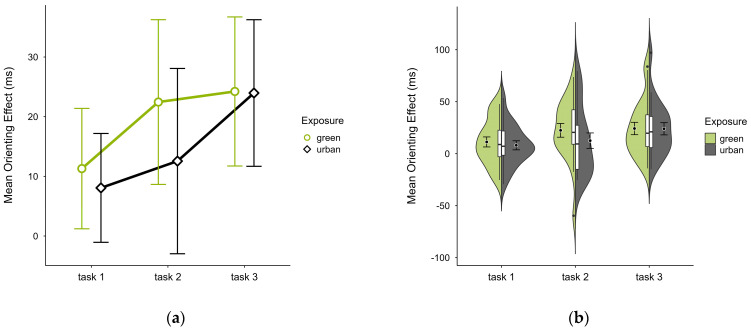
ANT orienting effect (ms) across task blocks and design conditions (green vs. urban): (**a**) mean changes over time with CIs; (**b**) distribution and variability of observations across task blocks, medians with IQRs (centre), means, and CIs (periphery).

**Figure 8 brainsci-15-00432-f008:**
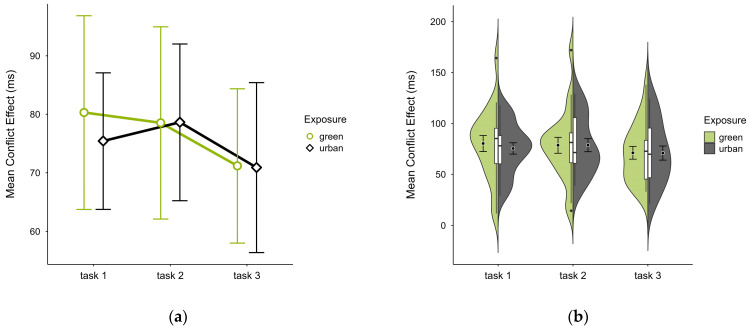
ANT conflict effect (ms) across task blocks and design conditions (green vs. urban): (**a**) mean changes over time with CIs; (**b**) distribution and variability of observations across task blocks, medians with IQRs (centre), means, and CIs (periphery).

**Figure 9 brainsci-15-00432-f009:**
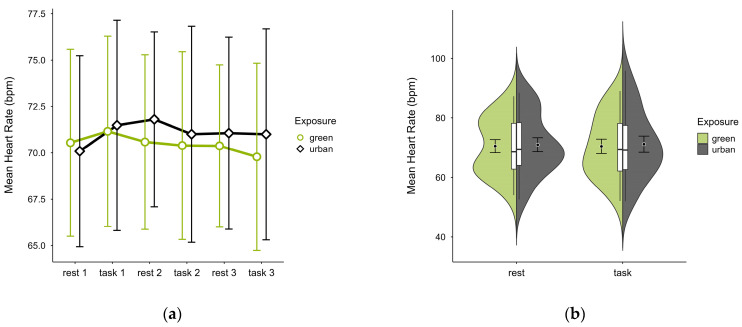
Heart rate (bpm) across trial blocks (rest, task) and design conditions (green vs. urban): (**a**) mean changes over time with CIs; (**b**) distribution and variability of observations across task blocks, medians with IQRs (centre), means, and CIs (periphery).

**Figure 10 brainsci-15-00432-f010:**
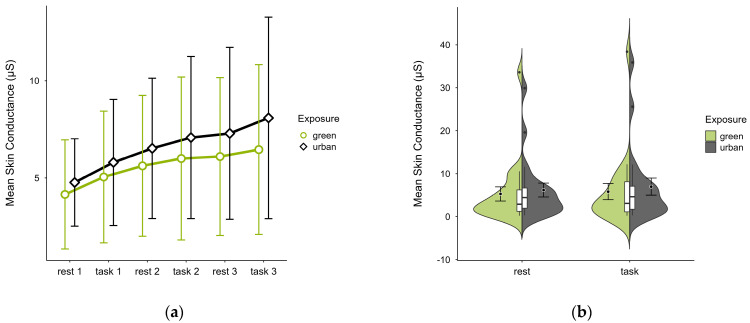
Skin conductance (μS) trial blocks (rest, task) and design conditions (green vs. urban): (**a**) mean changes over time with CIs; (**b**) distribution and variability of observations across task blocks, medians with IQRs (centre), means, and CIs (periphery).

**Figure 11 brainsci-15-00432-f011:**
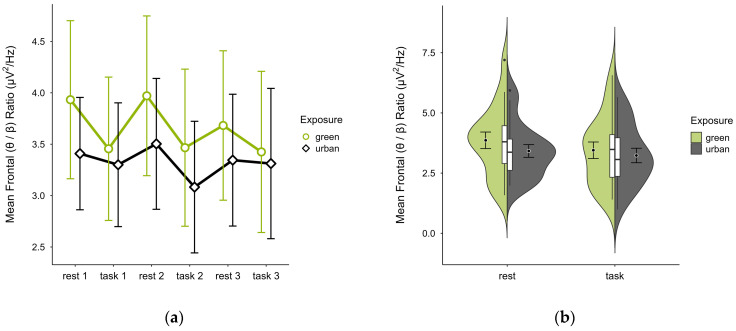
Frontal (θ/β) ratio across trial blocks (rest, task) and design conditions (green vs. urban): (**a**) mean changes over time with CIs; (**b**) distribution and variability of observations across task blocks, medians with IQRs (centre), means, and CIs (periphery).

**Figure 12 brainsci-15-00432-f012:**
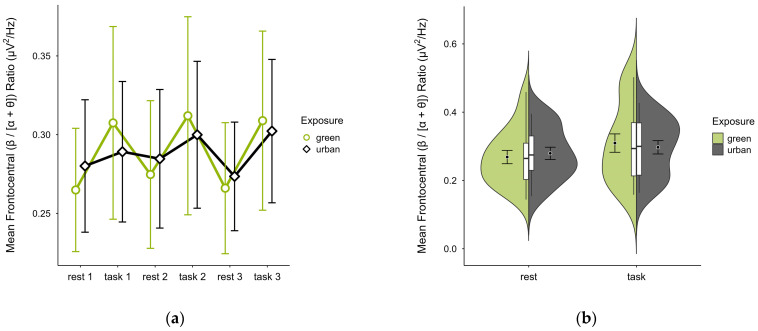
Frontocentral (β/[α + θ]) ratio across trial blocks (rest, task) and design conditions (green vs. urban): (**a**) mean changes over time with CIs; (**b**) distribution and variability of observations across task blocks, medians with IQRs (centre), means, and CIs (periphery).

**Figure 13 brainsci-15-00432-f013:**
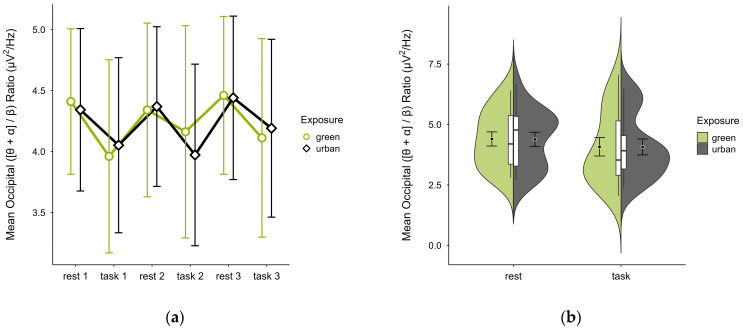
Occipital ([θ + α]/β) ratio across trial blocks (rest, task) and design conditions (green vs. urban): (**a**) mean changes over time with CIs; (**b**) distribution and variability of observations across task blocks, medians with IQRs (centre), means, and CIs (periphery).

**Figure 14 brainsci-15-00432-f014:**
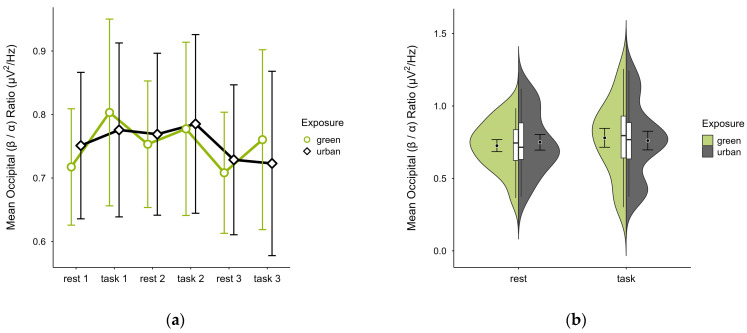
Occipital (β/α) ratio across trial blocks (rest, task) and design conditions (green vs. urban): (**a**) mean changes over time with CIs; (**b**) distribution and variability of observations across task blocks, medians with IQRs (centre), means, and CIs (periphery).

**Figure 15 brainsci-15-00432-f015:**
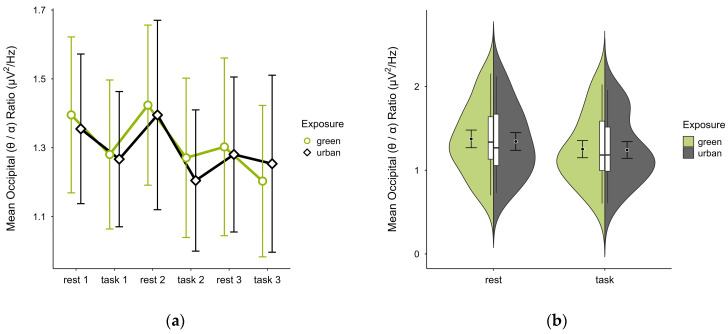
Occipital (θ/α) ratio across trial blocks (rest, task) and design conditions (green vs. urban): (**a**) mean changes over time with CIs; (**b**) distribution and variability of observations across task blocks, medians with IQRs (centre), means, and CIs (periphery).

**Table 1 brainsci-15-00432-t001:** Overview of regions of interest, selected EEG metrics, and research background.

ROI ^1^	Ch ^2^	Ratio	Dynamics	State	Interpretation	Ref. ^3^
frontal	F3, F4	θ/β	↓	task	attentional control	[30,31,32]
frontal	F3, F4	θ/β	↑	rest	mind wandering	[33]
frontocentral	FC5, FC6	β/[α + θ]	↑	task	engagement	[27,28,29,44]
occipital	O1, O2	[θ + α]/β	↑	task	fatigue	[36,37,38,39]
occipital	O1, O2	β/α	↑	task	arousal/stress	[34,35]
occipital	O1, O2	β/α	↓	task	fatigue	[37]
occipital	O1, O2	θ/α	↑	task	visual attention	[45]

^1^ ROI: region of interest; ^2^ Ch: channels; ^3^ Ref.: references of background research; ↓ decreased EEG ratio, ↑ increased EEG ratio.

**Table 2 brainsci-15-00432-t002:** Crossover group differences.

	N	Group G × U	n_1_	Group U × G	n_2_	*p*	Test ^2^
Female (n)	20	5/10	10	5/10	10	1.000	χ^2^
Age (years)	20	40.10 (15.33)	10	35.80 (13.90)	10	0.326	W
Washout period (days)	20	19.60 (5.80)	10	19.60 (7.86)	10	1.000	t
Δ Session start times (min)	20	75.55 (70.46)	10	86.07 (66.16)	10	0.853	W
Warm season sessions (n)	20	16/20	10	15/20	10	0.871	χ^2^
DSST processing rate (%) ^1^	20	41.79 (8.38)	10	45.18 (8.19)	10	0.372	t
DSST correct responses (%) ^1^	20	99.56 (0.75)	10	99.94 (0.21)	10	0.234	W
ANT reaction time (ms) ^1^	20	475.63 (66.42)	10	441.26 (43.81)	10	0.191	t
ANT correct responses (%) ^1^	20	97.88 (2.29)	10	97.95 (2.50)	10	0.449	W
ANT alerting effect (ms) ^1^	20	60.07 (19.57)	10	37.32 (15.13)	10	0.010 **	t
ANT orienting effect (ms) ^1^	20	18.63 (12.45)	10	15.56 (14.14)	10	0.612	t
ANT conflict effect (ms) ^1^	20	74.97 (28.00)	10	76.69 (17.39)	10	0.871	t
Heart rate (bpm) ^1^	19	68.10 (10.95)	9	73.18 (8.62)	10	0.282	t
Skin conductance (μS) ^1^	20	5.78 (8.14)	10	5.85 (5.17)	10	0.631	W
EEG F (θ/β) ratio (μV^2^/Hz) ^1^	18	2.74 (0.90)	9	4.25 (1.00)	9	0.004 **	W
EEG FC (β/[α + θ]) ratio (μV^2^/Hz) ^1^	18	0.33 (0.09)	8	0.26 (0.07)	10	0.103	t
EEG O ([θ + α]/β) ratio (μV^2^/Hz) ^1^	16	3.86 (1.44)	7	4.60 (1.00)	9	0.275	t
EEG O (β/α) ratio (μV^2^/Hz) ^1^	16	0.82 (0.26)	7	0.70 (0.18)	9	0.310	t
EEG O (θ/α) ratio (μV^2^/Hz) ^1^	16	1.23 (0.43)	7	1.36 (0.34)	9	0.498	t

Means and standard deviations. ^1^ Aggregated data across blocks and expositions. ^2^ Tests: χ^2^, Chi-Square test; W, Wilcoxon rank-sum test; t, Student’s *t*-test; F, frontal; FC, frontocentral; O, occipital. ** *p* < 0.01.

## Data Availability

The original contributions presented in this study are included in the Appendix A. Further inquiries can be directed to the corresponding author(s).

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
