# Peer review of "Neurophysiological Markers of Design-Induced Cognitive Changes: A Feasibility Study with Consumer-Grade Mobile EEG"

_brainsci, 2025, doi:10.3390/brainsci15050432_

Round 1
Reviewer 1 Report
Comments and Suggestions for Authors
This study assesses the plausibility of neurophysiological biomarkers of design induced cognitive changes. I find the idea interesting, however numerous aspects need to be clarified/fixed.
Comments:
Were the participants suffering from depression? What was the educational level of participants? Please clarify these items.
Did the authors applied a questionnaire to assess subject preference to experimental settings?
Was the testing room sound-proof?
Was the fatigue induction task employed at the beginning of each task block? How long was this fatigue induction?
Please clarify, how did you positioned the emotive cap on each subject and particularly indicate the approach you followed to secure proper placement of electrode positions?
Please clarify, how did you remove blink and muscle artifacts on the EEG?
Please clarify the parameters of the low 5 μV2 /Hz low-pass filter?
Please provide the percentage of data that was discarded because of artifacts.
You stated “EEG metrics were selected based on a review of available evidence.” Please provide the reference.
Please provide a graph of EEG spectral power for channels (F3, F4, O1, O2, FC5, FC6) across subjects before and after artifact removal.
Please provide a graph of EEG spectral power for channels (F3, F4, O1, O2, FC5, FC6) across subjects for the conditions green vs. urban.
The authors indicated that groups showed a significant difference in ANT alerting effect and EEG frontal (θ / β) ratio. Please discuss reasons for these differences?
Regarding skin conductance, please specify what parameter did you compare?
Please provide a statement about the availability of the data for other researchers.
Author Response
We would like to thank the reviewer for the time taken to read our manuscript so thoroughly and bringing up all these important questions. Please find our detailed responses and the corresponding changes in the attached PDF file.

Reviewer 2 Report
Comments and Suggestions for Authors
In the present study the authors investigated the effects on various EEG power band ratios of two different evidence-based interior designs (nature-inspired vs urban-inspired).
The study has many limitations. Specifically, the analyzed sample includes only 20 participants, despite the absence of strong exclusion/inclusion criteria that may complicate the enrolment procedure. Furthermore, among the considered EEG features, only the frontal (θ / β) ratio shows statistically significant differences between the two conditions.
However, the paper is well written and the analysis is conducted correctly. Furthermore, designing and conducting naturalistic EEG experiments is still a challenging task and this paper presents a possible experimental procedure to solve this task in the context of evidence-based design. In this sense, I found the paper interesting for the audience of the journal and I therefore recommend its publication after that the authors have considered the following points.
- Even though the use of evidence-based design should be the core of this work, to me it is not clear the role in the experiment of the experimental stimuli (Figure b). Were the subjects instructed to look at them when performing the task or were they simply positioned in the room? Why is a neutral stimulus needed?
- I think it would be interesting to compare the average EEG spectra in the considered bands (theta, alpha, low/high beta) before comparing the ratios in Table 1.
- It would be interesting to verify to what extent results change when individually-defined frequency bands are used instead of the standard one (see Vallarino et al. Human Brain Mapping, 2022). I do not expect the authors to re-run the whole analysis by replacing the definition of frequency bands, but at least they should add a comment on this.
- Why was task performance during the phase of fatigue induction not analyzed?
- Please, increase the font size of axis labels and legends in all figures so as to make them readable.
- Line 75, I am not sure that the expression ‘proprietary EEG metric’ is correct
Author Response

(The authors gave the same response as above.)
